# Sustainable Management of Calcite Contaminated with Waste Toner Powder in the Construction Industry

**DOI:** 10.3390/ma15144785

**Published:** 2022-07-08

**Authors:** Halyna Kominko, Piotr Radomski, Anna K. Nowak, Zbigniew Wzorek

**Affiliations:** 1Department of Chemical Technology and Environmental Analytics, Faculty of Chemical Engineering and Technology, Cracow University of Technology, Warszawska Street 24, 31-155 Cracow, Poland; piotr.radomski@pk.edu.pl (P.R.); zbigniew.wzorek@pk.edu.pl (Z.W.); 2Department of General and Inorganic Chemistry, Faculty of Chemical Engineering and Technology, Cracow University of Technology, Warszawska Street 24, 31-155 Cracow, Poland; anna.k.nowak@pk.edu.pl

**Keywords:** waste toner powder, concrete, calcite, sustainable management

## Abstract

Due to the highly explosive nature of toners, absorbers are used in toner processing plants to prevent the explosion of toner dust suspension in the air. Usually, finely divided calcite (in the form of a dust) is used. The mixture of toner-calcite is treated as waste and landfilled. The main aim of this study was to investigate the possibility of using toner-contaminated calcite as an additive to concrete. Materials originating from the toner processing plant were analyzed by using TGA, AAS, XRD, FTIR and SEM techniques. Calcite-waste toner powder mixture in amounts 0%, 1%, 5%, 10%, 15% and 20% were used to produce concrete. The results of the study showed that an increase in the amount of calcite contaminated with toner to 20% causes a decrease in compressive strength of concrete by 24–51% depending on material sample. The addition of calcite in amounts up to 5% can be a suitable method of its management.

## 1. Introduction

With rapidly changing technologies and constant consumer demand for the latest devices, the amount of e-waste is constantly growing. Used cartridges constitute a significant part of this waste and their estimated amount is 7200 tons [1]. This type of waste still contains toner (near 10% [2]), and if managed in an inappropriate way, it poses a threat to the environment and humans. 

Toners usually contain a resin binder (polyethylene, polypropylene, polystyrene, epoxy resins, acrylics, etc.), colorant (dyes and pigments), charge control agent (hydroxyl-aromatic acid and the derivatives), releasing agent and other additives (colloidal silica, metal salts, metal salts of fatty acids) [3,4]. It is estimated that only 20–30% of toners is recycled globally [2]. The rest are usually landfilled and can cause soil, water pollution and, what is more, lead to serious respiratory problems in humans due to small fine particles (5–10 μm) [5]. 

Increasing the level of recycling of used toners is an important issue not only from the environment point of view, but also resource and energy conservation and economics. A lot of research has been conducted in order to find alternative use for waste toner. Ruan and co-workers [3] proposed a mechanical production line for magnetic separation of steels and magnets from aluminum and plastics. The efficiency of the process reached >95% and recovered materials can be used as secondary materials. Waste toner can also be successfully used as an enhancement modifier in asphalt binders and mixtures [6,7]. Kumar and co-workers [8] conducted research on possibilities of recovering iron from used toner. An interesting application is presented in [9], where waste toner in combination with chitosan is used as a carrier for enzymes. Study [10] armed to evaluate the potential of waste toner use in the design of microwave absorbers. Absorbing properties of used toner was also verified in removal of cadmium and chromium (VI) [11,12]. 

A promising solution to the problem of waste toner disposal may be its management in the construction industry. Cement production is incredibly energy intensive and one of the main emitters of carbon dioxide emissions. According to the estimation, production of 1 tonne of cement releases approximately 0.9 tonnes of CO_2_, which is 7% of the total CO_2_ global emission [13]. Therefore, the use of pozzolanic industrial waste and by-products [14,15,16] as cement replacement materials can be considered as a sustainable solution in order to reduce the amount of cement in concrete mixture and consequently reduce CO_2_ emissions [17]. 

Newland and co-workers [18] investigated the possibility of using waste toner powder as a pigment in concrete. The results of study showed that replacement of cement with toner at the level 5–10% does not impact significantly on selected properties of concrete. Waste toner powder can also be used as an additive to foamed concrete. According to [19], the addition of toner (5%) to concrete increases the compressive strength of the final product by about 30%. The above-mentioned studies concern the impact of waste toner powder on the properties of concrete. The authors did not find publications that dealt with a real mixture of calcite and toner as waste arising during the processing of used printers. The interest in this topic results from the fact that in Poland, such a mixture is stored, and toner processing plants incur high costs related to the landfill of this waste.

The aim of the present study was to characterize calcite contaminated with waste toner powder and assess a possibility of its management in the construction industry. The calcite–toner mixture was characterized by using XRD, FTIR, AAS, and SEM-EDS techniques. Additionally, then the influence of a different amount of the addition of this mixture on the properties of the cement was assessed. 

## 2. Materials and Methods

### 2.1. Samples

The test samples came from a toner processing plant. When separating the toner powder from ferrous and non-ferrous metals and plastics, the toner powder is diluted with ground calcite to avoid an explosion. Obtained mixture is waste, which is landfilled. 

### 2.2. X-ray Analysis of Samples

To determine the crystal structure of the samples, the X-ray diffraction (XRD) technique was used. The analysis was performed on a Philips X’Pert apparatus (PANalytical, Almelo, Netherlands) equipped with a graphite monochromator PW 1572/00 Cu Kα (λ = 0.15418 nm) with a Ni filter (40 kV and 40 mA). The diffractograms were registered in the range of diffraction angles 2θ in the range of values from 10° to 60°, using a step size of 0.025°. The phase composition of the materials was identified on the basis of the PDF-2 ICDD (the International Centre for Diffraction Data) database. Crystallite size was determined using a line position and line shape standard NIST 660c (lanthanum hexaboride powder).

### 2.3. FTIR Test of Calcite Contaminated with Waste Toner

To characterize the functional groups in calcite contaminated with waste toner powder, the FTIR (Fourier-transform infrared) analysis was performed. The measurements were conducted with a Nicolet iS5 Thermo Scientific spectrophotometer (Thermo Fisher Scientific, Waltham, MA, USA) equipped with an iD7 diamond ATR (Attenuated Total Reflection) accessory. Spectra at a resolution of one data point per 1 cm^−1^ were obtained with wave numbers from 4000 to 400 cm^−1^ at room conditions. The spectra were obtained with respect to a background, which has been taken of the air under the same measurement conditions.

### 2.4. Thermogravimetric Analysis of Samples (TGA)

Thermal analysis of the samples was carried out with the use of EXSTAR SII TG/DTA 7300 apparatus. The tests were carried out in a platinum crucible, in the temperature range of 30 °C–1000 °C, at a temperature increase of 20 °C/min. The analysis was performed in an air atmosphere with a flow of 200 mL/min.

### 2.5. Testing of the Elemental Composition Using the AAS Method

Chemical composition of the tested materials was determined by using Atomic Absorption Spectroscopy (AAS) after digestion in a mixture of concentrated nitric and chloric acids (3:1 (*v*:*v*)). The determination of the content of selected elements in the solutions after mineralization was carried out on a Perkin Elmer 370 apparatus (Perkin Elmer, Waltham, MA, USA) in duplicate for each sample. For the preparation of standard solutions for calibration curves the standards from Sigma Aldrich were used.

### 2.6. Observation of the Sample Surface Using the SEM Method

Morphological observations on the samples were performed using the Apreo S LoVac Scanning Electron Microscope (SEM) (Thermo Fisher Scientific, Waltham, MA, USA) equipped with a low-vacuum backscattered electron detector (BSE) and energy dispersion X-ray microanalyzer (EDS).

### 2.7. Strength Tests

B-25 concrete mixture was prepared for strength tests (according to [20]). One part of the certified Portland cement was mixed with half a part of water and then, with continued stirring, two parts of standard sand were added. Obtained mixture was divided into 6 parts, and the following amount of calcite contaminated with waste toner powder was added to each one: 0%, 1%, 5%, 10%, 15%, 20% of total weight of the mixture. The prepared mixtures were transferred to the molds of 40 × 40 × 160 mm in order to obtain beams. Two beams were made for each calcite addition. After one day, the beams were placed in a water bath for 24 h. After this, the samples were removed from the molds and transferred to maturation for 28 days. During this time, the obtained beams were cared for by wetting and overturning. Then, strength tests were carried out on an universal machine for standardized strength tests, mainly of ceramic and building materials Zwick/Roell Z050. The speed of lowering the pressure on the beams was 0.01 mm/s. In the case of breaking strength test the three-point method of loading was used, which consisted in placing a beam in the machine with the side surface on the support rollers. Then, by means of a load roller, the load was transferred vertically and the pressure was evenly increased until the tested beam was broken. In order to carry out the comprehensive strength test, the beams were placed with the side surface in the center of the plate in the longitudinal direction, then its side surface was loaded and increased until the beam was crushed.

## 3. Results

### 3.1. The Phase Composition of Calcite Contaminated with Waste Toner Powder

The results of XRD analysis are presented in Figure 1. All calcite samples have the same diffractogram pattern. Tested materials displayed reflections at 23°, 29°, 31°, 35°, 39°, 43°, 47°, 48° and 58° 2θ, which confirms the presence of calcium carbonate. The similar diffractogram of calcite was obtained in [21]. In all samples calcite crystalizes in the trigonal system. The hexagonal unit cell for sample A has a = b = 4.9803 Å and c = 17.0187 Å. In case of samples B and C, the unit cell is characterized by the same dimensions a = b = 4.9890 Å and c = 17.0620 Å. Degree of crystallinity in all samples of calcite contaminated with waste toner powder is about 97%. The calcite crystallite size was 654 ± 103 Å, 543 ± 69 Å and 743 ± 153 Å for sample A, B and C, respectively.

### 3.2. The Characterization of Functional Groups

As can be seen from Figure 2 the samples of calcite contaminated with waste toner powder have a similar FTIR spectra. The υ_4_ band (symmetric CO_3_ deformation) located at 712 cm^−1^ in samples A and B and υ_2_ band (asymmetric CO_3_ deformation) at 872 cm^−1^ in all samples indicate the presence of calcite [22]. The peak, which appeared at 1394 cm^−1^, can also correspond to an asymmetric strength band of calcite. In the case of sample C, the peak at 1794 cm^−1^ may be regarded as the combination band of υ_1_ + υ_4_ of carbonate ion [23]. The absorption band at 697 cm^−1^ in all tested materials can represent Si-O-Si stretching vibrations [24].

### 3.3. The Results of TGA Analysis

The results of TGA analysis are presented in Figure 3. In all samples, a decrease in weight of 5.4–6.9% from the initial value was observed. This loss occurs in the temperature range of 200 °C–420 °C and is accompanied by the release of heat. This proves the presence of organic material in the analyzed material (toner), which melts and burns in this temperature range [8]. After the mass stabilizes and the temperature increases further, decomposition of calcium carbonate occurs. Due to the significant fragmentation of the analyzed materials, it occurs at temperatures above 700 °C and is accompanied by an endothermic effect [25]. In all analyzed samples, this loss was 41–42%, which confirms that the dominant component of the materials is calcium carbonate (for pure material this loss is 44% [26]). Based on the TGA analysis, it can be concluded that the tested material contains no more than 7% of toner.

### 3.4. Chemical Composition

The content of selected elements in calcite contaminated with waste toner powder is presented in Table 1. The tested materials are characterized by very similar chemical composition. The content of Ca varied from 25.8% to 27.5%, the Mg content was in the range of 0.13–0.14%. The content of other analyzed elements was below the limit of quantification. In terms of heavy metals, calcite contaminated with waste toner powder is safe for the environment. 

### 3.5. The Results of the Surface Observation 

In the SEM micrographs (Figure 4, Figure 5 and Figure 6), apart from CaCO_3_ crystallites, scattered particles of similar size and clearly darker (more contrasting) color are visible. EDS analysis of these particles presented in Table 2, Table 3, Table 4 and Table 5 showed that the dominant component is carbon, which is the main element in the organic materials used for the production of printing toners (mainly in the form of a polymeric material). Silica was identified in samples A and C of calcite contaminated with waste toner powder. It was also confirmed by the FTIR analysis. Cu, Mg and Fe were found in sample C, which can be considered as impurities.

### 3.6. Strength Tests

Figure 7 presents a picture of concrete beams used for strength tests, the results of which are presented in Table 5 and Table 6. The addition of calcite contaminated with waste toner powder to concrete mixture results in a decrease in compressive strength of concrete. Compressive strength decreases with increasing calcite content. The exception is a beam of 10% calcite addition, where the value of compressive strength (26.9 MPa) is similar to the value for control beam without waste material (27.1 MPa) (Table 5). The highest compressive strength was obtained for the concrete with the addition of sample A. The concrete with calcite contaminated with waste toner powder in the amount of 20% caused a decrease in compressive strength by 22% compared to the control sample. The addition of sample B of calcite resulted in a decrease in compressive strength of concrete by 24% in the case of 1% calcite and by 44% in the case of 20% calcite. The concrete with the addition of sample C showed a decrease in strength by 14% and 51% with the content of 1% and 20% calcite, respectively. 

Besides the fact that calcite is not characterized by pozzolanic properties, it can have chemical effects or physical effects (nucleation, filler, dilution) on concrete mixture. This effect depends on the amount of calcite and degree of fragmentation [27,28]. Calcite can react with aluminate phases in hydrating cement to form carboaluminate phases, among which mono- and hemicarboaluminate are the most common. This may result in an increase in comprehensive strength of cement [29]. However, if not all of the carbonate is used to form the carboaluminate it can be unstable, which will have a negative effect on the mechanical properties of the cement [30]. Supit and co-workers [31] performed a study to determine the impact of nano-CaCO_3_ on concrete properties. The results showed that the increase in nano-CaCO_3_ content from 1% to 4% decreased the comprehensive strength of mortars, which can be attributed to the agglomeration of nano-CaCO_3_ due to higher van der Waal’s forces than in cement. In study [32] a mixture of waste toner and calcite (1:1 (wt)) was added to the concrete in amounts of 1%, 3%, 5% and 10%. The results of the research showed that the samples with 1%, 3% and 5% replacement did not differ considerably from the reference sample. 

**Figure 7 materials-15-04785-f007:**
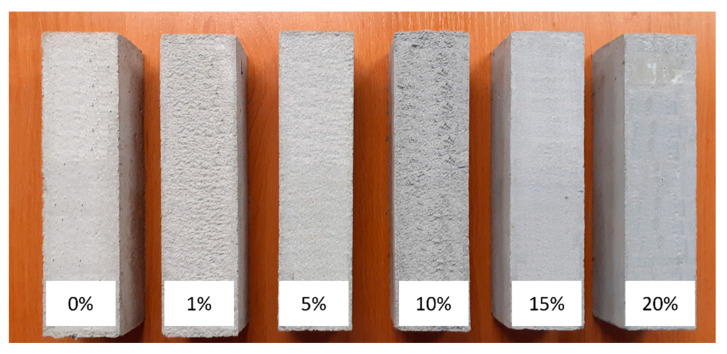
Pictures of concrete beams with the addition of calcite contaminated with waste powder toner (sample A).

**Table 5 materials-15-04785-t005:** Compressive strength of concrete with the addition of calcite contaminated with waste toner powder.

	Compressive Strength, MPa					
	Content of Calcite Contaminated with Waste Toner Powder in Concrete Mixture, %					
	0	1	5	10	15	20
Sample A	27.1 ± 0.2	25.3 ± 0	24.4 ± 0	26.9 ± 0.3	22.1 ± 0.6	21.2 ± 0.2
Sample B		20.5 ± 0.6	19.2 ± 0.1	19.1 ± 0.3	17.5 ± 0.4	15.1 ± 0.1
Sample C		22.2 ± 0.1	22.7 ± 0.4	20.9 ± 0.3	14.3 ± 0.1	13.3 ± 0.2

**Table 6 materials-15-04785-t006:** Breaking strength of concrete with the addition of calcite contaminated with waste toner powder.

	Breaking Strength, N					
	Content of Calcite Contaminated with Waste Toner Powder in Concrete Mixture, %					
	0	1	5	10	15	20
Sample A	1480 ± 28	1280 ± 57	1660 ± 14	1530 ± 14	1280 ± 42	1210 ± 14
Sample B		1220 ± 113	1330 ± 28	1230 ± 28	1120 ± 14	913 ± 46
Sample C		1450 ± 14	1530 ± 42	1360 ± 14	990 ± 14	782 ± 25

In general, breaking strength of concrete decreases with the addition of calcite contaminated with waste toner powder (Table 6). In the case of sample A, in the amount of 5% and 10%, the values of breaking strength of concrete were higher than control sample. A breaking strength increase was also observed for the concrete with the addition of 5% of sample C in comparison with control. This may be related to non-homogeneous mixing of the material or inappropriate compaction of the concrete sample after entering the mold. With the increase in the amount of calcite introduced into the concrete, changes in the consistency of the concrete were observed, the higher the amount of calcite introduced, the denser the mortar became (due to the highly hygroscopic nature of the waste), which could lead to inappropriate compaction. 

The best results were obtained for the concrete with the addition of sample A of calcite. The decrease in breaking strength with 20% calcite content in concrete was 18%. In the case of samples B and C, the decrease in breaking strength was 18–38% and 2–47%, respectively, depending on the calcite content. 

## 4. Discussion

Based on the research results, it can be concluded that the analyzed material, apart from calcium carbonate in the calcite form, contains no more than 7% of printing toner powder. The addition of this material (regardless of the sample), up to 5%, did not cause a significant change in the breaking strength, only the introduction of a larger amount resulted in a decrease in this value. The same was observed in the case of the compressive strength test. There were quite noticeable differences in compressive strength between the individual samples of calcite contaminated with waste toner powder. These discrepancies may result from poor uniformity of the samples or poor compaction during the concrete beams preparation. Regardless of that, in all compressive strength tests, the obtained values of the compressive strength were lower than the pure concrete without the addition of calcite.

The addition of calcite contaminated with waste toner powder to concrete materials is a potential direction in the management of this type of waste. In order to determine the wider use of concrete composites with the addition of calcite contaminated with waste toner powder, it is necessary to perform additional detailed analyses. These analyses should include the influence of the additive on the setting time of the slurry, water absorption, resistance to weather conditions and the influence of the setting time on the mechanical strength of binders. Conducting these analyses will allow the target development direction along with the optimal composition for concrete composites with the addition of calcite.

## 5. Conclusions

The waste from a toner processing plant, which is calcite containing up to 7% of toner powder, can be used as an additive to the concrete. The introduction of a small amount of this material (<5%) into concrete, as shown by the tests carried out, does not significantly affect one of the key application parameters related to strength. The use of calcite contaminated with waste toner powder for concrete mixes will reduce its storage and further processing, which is quite problematic due to the completely different chemical properties of toners compared to calcite. The obtained composites or ready-to-use loose mixes with concrete can be used in the production of non-structural concrete or as structural and reinforcing materials for the foundation, stabilization or protection of road landfills, etc.

## Figures and Tables

**Figure 1 materials-15-04785-f001:**
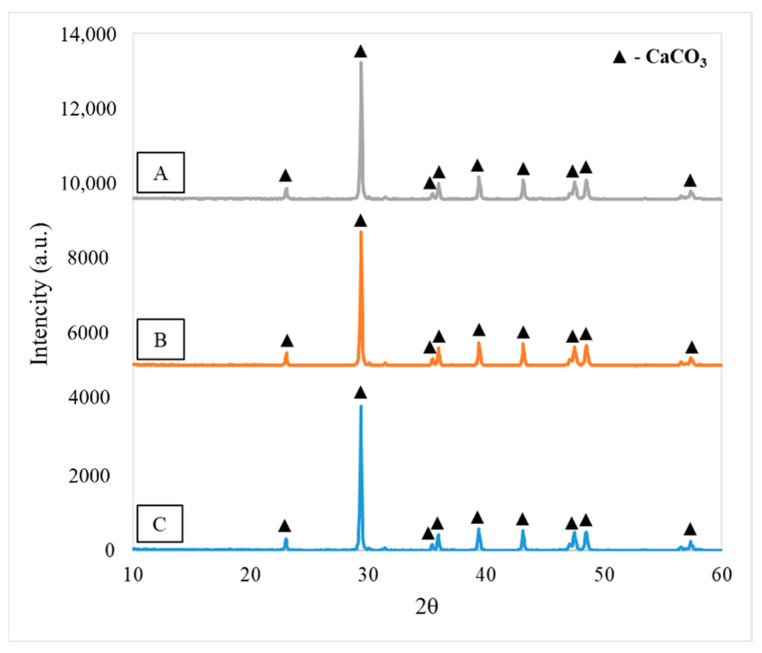
Diffractogram patterns of tested materials ((**A**)—sample A, (**B**)—sample B, (**C**)—sample C).

**Figure 2 materials-15-04785-f002:**
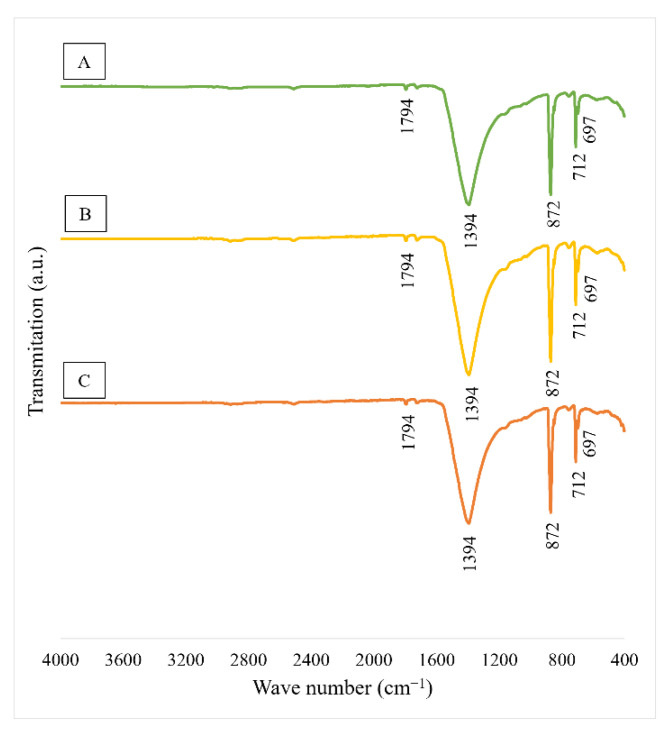
FT-IR spectra of tested materials ((**A**)—sample A, (**B**)—sample B, (**C**)—sample C).

**Figure 3 materials-15-04785-f003:**
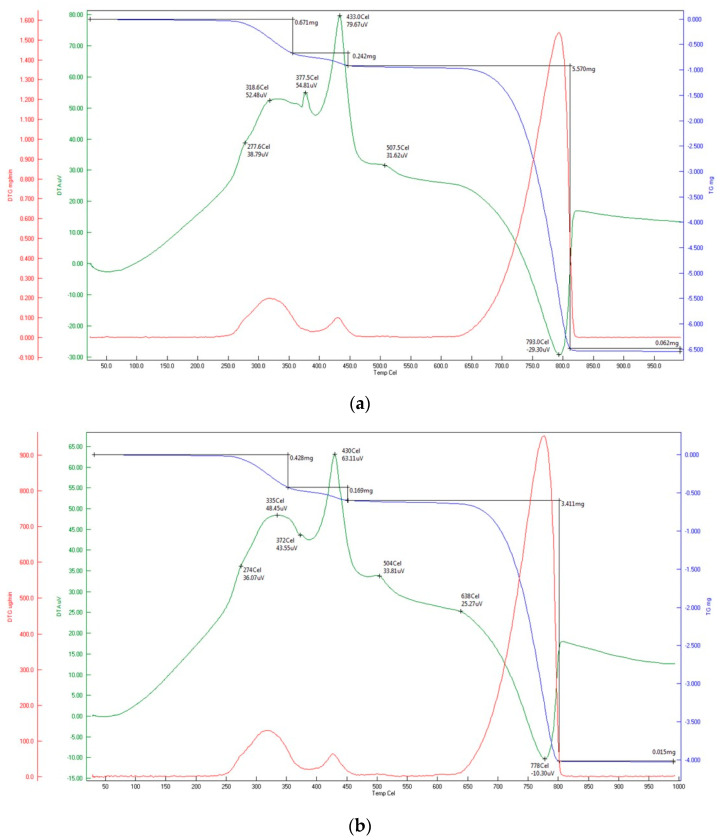
Results of TGA analysis: (**a**) sample A, (**b**) sample B, (**c**) sample C.

**Figure 4 materials-15-04785-f004:**
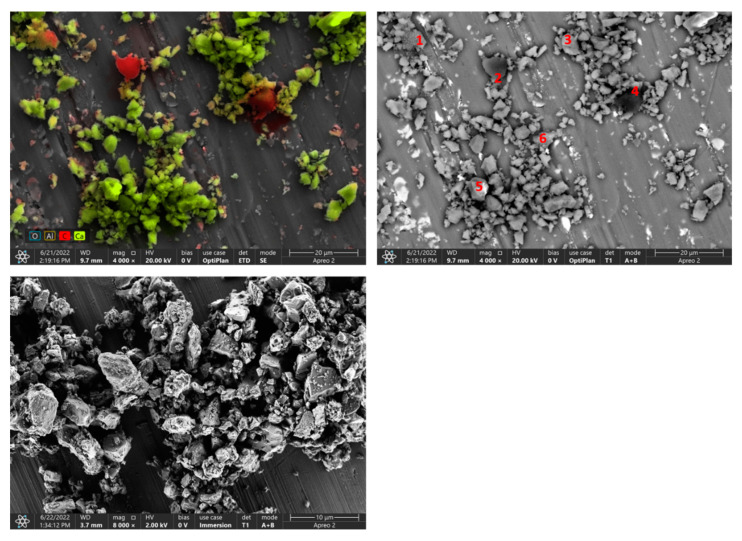
SEM-photographs and EDS analysis of sample A of calcite contaminated with waste toner powder (red numbers mark EDS analysis points).

**Figure 5 materials-15-04785-f005:**
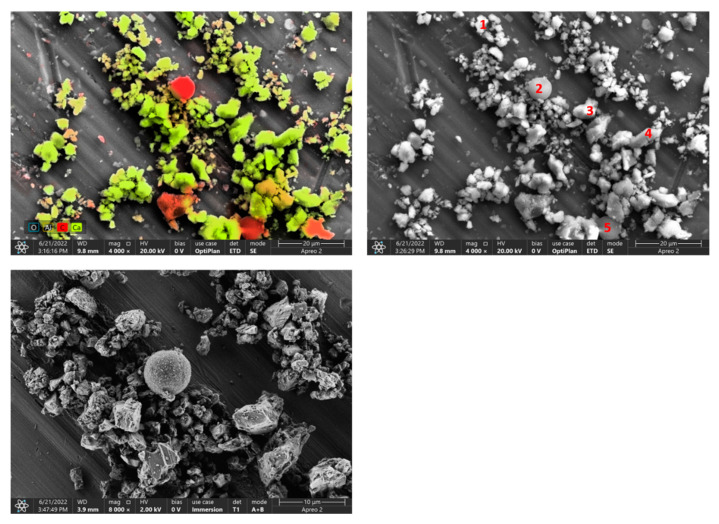
SEM-photographs and EDS analysis of sample B of calcite contaminated with waste toner powder (red numbers mark EDS analysis points).

**Figure 6 materials-15-04785-f006:**
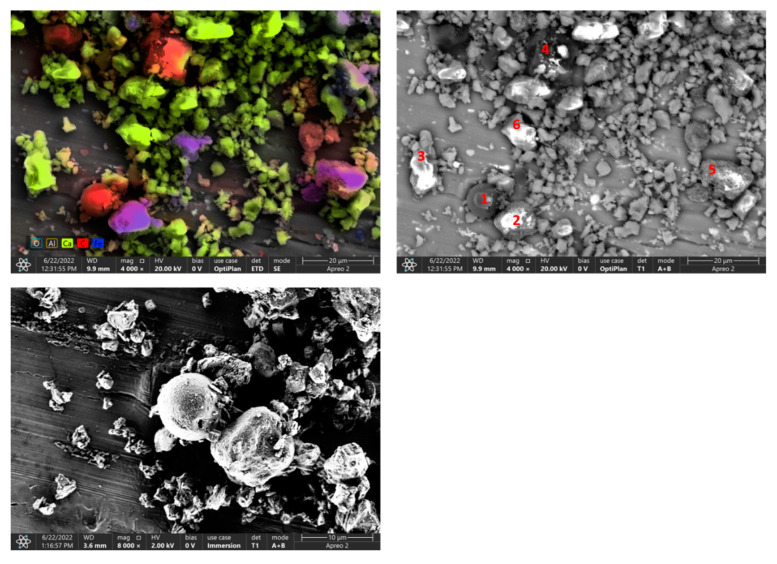
SEM-photographs and EDS analysis of sample C of calcite contaminated with waste toner powder (red numbers mark EDS analysis points).

**Table 1 materials-15-04785-t001:** Content of selected elements in calcite samples.

		Sample A	Sample B	Sample C
Al	%	<0.004		
Ca		26.2	25.8	27.5
Cd		<0.01		
Cr		<0.007		
Cu		<0.004		
Fe		<0.001		
Mg		0.14	0.13	0.14
Mn		<0.016		
Ni		<0.001		
Pb		<0.004		
Si		<0.004		
Sn		<0.004		
Zn		<0.001		

**Table 2 materials-15-04785-t002:** Results of EDS analysis of sample A.

	C	O	Ca	Si	Fe
Spot 1					
Weight, %	58.7	32.6	0.9	0.3	8.6
Error, %	11.8	12.4	12.6	20.6	5.8
Spot 2					
Weight, %	82.1	17.3	0.6		
Error, %	10.4	14.0	0.2		
Spot 3					
Weight, %	20.6	46.5	32.9		
Error, %	16.7	12.8	3.3		
Spot 4					
Weight, %	87.9	11.0	1.0		
Error, %	10.6	19.0	14.6		
Spot 5					
Weight, %	26.1	54.9	18.9		
Error, %	14.1	11.9	3.3		
Spot 6					
Weight, %	16.9	55.1	28.0		
Error, %	18.7	12.7	3.5		

**Table 3 materials-15-04785-t003:** Results of EDS analysis of sample B.

	C	O	Ca
Spot 1			
Weight, %	31.3	44.4	23.3
Error, %	15.5	12.5	3.5
Spot 2			
Weight, %	86.3	13.1	0.7
Error, %	10.3	14.4	13.3
Spot 3			
Weight, %	27.3	54.1	18.5
Error, %	14.1	11.8	3.3
Spot 4			
Weight, %	22.5	50.3	27.2
Error, %	15.7	12.4	3.3
Spot 5			
Weight, %	86.9	12.3	0.9
Error, %	10.1	14.9	14.2

**Table 4 materials-15-04785-t004:** Results of EDS analysis of sample C.

	C	O	Ca	Si	Fe	Cu	Mg
Spot 1							
Weight, %	94.3	5.1	0.5	0.1			
Error, %	9.3	3.9	14.5	32.4			
Spot 2							
Weight, %	74.5	14.3		0.2	10.9		
Error, %	10.7	13.9		0.1	4.3		
Spot 3							
Weight, %	28.4	42.2	35.0			7.9	
Error, %	23.9	15.3	7.4			10.6	
Spot 4							
Weight, %	96.8	2.5			0.7		
Error, %	9.7	29.5			26.1		
Spot 5							
Weight, %	65.9	20.2		0.4	13.2		
Error, %	11.0	13.0		16.2	3.8		
Spot 6							
Weight, %	28.0	55.0	16.2				0.3
Error, %	12.2	11.2	2.7				12.0

## Data Availability

Not applicable.

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
