# Peer review of "Sustainable Management of Calcite Contaminated with Waste Toner Powder in the Construction Industry"

_materials, 2022, doi:10.3390/ma15144785_

Round 1

Reviewer 1 Report

1. The absorption band at 697 cm-1 is strong, does it mean it contains a certain amount of silica.

2. The sample contains about 7% toner, but this article lacks the analysis of the influence of toner on the mechanical properties of concrete.

3. From Fig.4a, scattered particles of similar size and clearly darker (more contrasting) color are visible”,does it toner, how to prove this point.

4. From Fig.4c, the dominant component of should be calcium, not carbon. And please provide the specific content of each element from the result of EDS.

5. “4. Discussion” should be 4 Conclusion.

Reviewer 2 Report

This manuscript characterizes waste toner-contaminated calcite through relevant tests and evaluates its management possibilities in the construction industry, some useful results were obtained. The manuscript requires major revision, and the possibility of acceptance is assessed after revision.

(1) The research background in the abstract should be more concise. Where is the significance of the research in the abstract? The abstract seems too long and should be limited to 150 words.

(2).” if be managed in an inappropriate way, pose a threat to the environment and humans”. There seems to be a lack of content about the aims of this study.

(3) “Toners usually contain resin binder …………small fine particles (5-10 μm) [5])” This section as a single paragraph seems to make the introduction better structured.

(4). The penultimate paragraph of the introduction should be based on the relevant literature, summarizing certain gaps in the literature that the author's team will investigate in this article.

(5). The last paragraph of the introduction lacks the research direction and concise research method of this study

(6). The introduction should be rewritten and give a clear research logic of this manuscript.

(7) Secondary headings are too simple and less readable. For example "FTIR" should be changed to "FTIR test of calcite contaminated with waste toner", Please revise the full text.

(8) “30-1000°C, at a temperature increase of 20°C/min” There should be a space between the value and the unit, Please check the full text.

(9) Section 2.7, the test process is not clear, the test process diagram should be supplemented to make the process easy to understand

(10) Sections 2.2 and 3.1 are both named XRD? I think it's a big question, and the title should distinguish between test methods and conclusions. Section 3.1 can be roughly changed to "The phase composition of calcite?" Of course, the author team can choose a more reasonable title. Please check the title for the full text.

(11) Figures 1 and 2 should be more concise, such as grid lines should be deleted, and the drawings should be poor and not aesthetically pleasing. Replace "intensity, a.u." with "intensity(a.u.)", please check full text image and axis labels.

(12)The similar diffractogram of calcite was obtained in [21]. This sentence is not necessary, if necessary, please explain.

(13)In the case of sample C, the peak at 1794 cm-1 may be regarded as the combination band of υ1+υ4 of carbonate ion [23]. The absorption band at 697 cm-1 in all tested materials can represent Si-O-Si stretching vibrations [24]This is the conclusion drawn by the author's team, why the reference is added? The following also has this problem, please check.

(14)5.35-6.91%and41.0-42.4% Expressing numerical values like this is not academic, Please check this issue in full.

(15) Section 3.6, why the 1% calcite content was tested, should be explained in more detail in Section 2.7.

(16) Strength testing was performed using a Zwick/Roell Z050 instrument, and what were the parameters at the time of testing, such as loading rate. It should be supplemented in more detail in Section 2.7.

(17) Section 4 “The management of waste toner is quite…………. in the next few years [33].” This paragraph should be a brief summary of the full-text research, please rewrite the paragraph, and no references should appear in the Discussion section.

(18)” as shown by the tests carried out,” Please specify what test.

Reviewer 3 Report

This study is characterized by a high level of originality and scientific soundness.

It is presented in a really appropriate way, both technically and scientifically. The conclusions are supported the quantity and quality of the data presented and by the results.

Reviewer 4 Report

Comments

This paper studied waste toner powder in the construction. The outcome of the paper is interesting however, there are several aspects that need to be improved. The reviewer can only recommend for publication if the author satisfactorily address the following major comments in the revised version.

1.       The research gap from the literature review should be clearly presented.

2.       The research questions and justification of selecting variables should be highlighted.

3.       Which test standards was considered in this study? How many replicate samples were tested in each category?

5.       The novelty of the study should be highlighted more clearly at the end of introduction section. How this study is different from the published study in literature?

6.       How the outcome of this study will benefit researchers and end users? This need to be highlighted in introduction or end of conclusion.

7.       The waste powder in construction is interesting but not novel. Therefore, the recent application in this area should be discussed in introduction section to improve the background study. Recently, the waste ceramic powder was used in composites [Ref: Effect of fire-retardant ceram powder on the properties of phenolic-based GFRP composites], and some other wastes were used in structural application [Ref: Investigation on the physical, mechanical and microstructural properties of epoxy polymer matrix with crumb rubber and short fibres for composite railway sleepers]. Suggest to include them in introduction section with proper citations to improve the background study.

I would be happy to see the revised version to understand how these comments are being addressed.

Round 2

Reviewer 2 Report

The authors have revised this manuscript hardly, I did not find some technical problem in this version. However, the explain for the figure 1 is too simple, it is not enougth to explain the phase composition of the calcite contaminated with waste toner powder, it must be improved extensively.

Author Response

Dear reviwer, thank you very much for your comments and remarks.

We added information about crystallite size of calcite, degree of crystallinity and dimension of unit cells to the manuscript.

Reviewer 4 Report

I have no further comments

Author Response

Dear reviewer, thank you for your comments and remarks.